# Early Intensive Physical Rehabilitation Combined with a Protocolized Decannulation Process in Tracheostomized Survivors from Severe COVID-19 Pneumonia with Chronic Critical Illness

**DOI:** 10.3390/jcm11133921

**Published:** 2022-07-05

**Authors:** Malcolm Lemyze, Matthieu Komorowski, Jihad Mallat, Clotilde Arumadura, Philippe Pauquet, Adrien Kos, Maxime Granier, Jean-Marie Grosbois

**Affiliations:** 1Department of Critical Care Medicine, Arras Hospital, 62000 Arras, France; cloaudegond@hotmail.com (C.A.); philippe.pauquet@gh-artoisternois.fr (P.P.); adrien.kos5962@gmail.com (A.K.); maxime.granier@gh-artoisternois.fr (M.G.); 2Department of Surgery and Cancer, Faculty of Medicine, Imperial College London, Exhibition Road, London SW7 2AZ, UK; matthieu.komorowski@gmail.com; 3Department of Critical Care Medicine, Critical Care Institute, Cleveland Clinic Abu Dhabi, Abu Dhabi 112412, United Arab Emirates; mallatjihad@gmail.com; 4Cleveland Clinic Lerner College of Medicine, Case Western Reserve University, Cleveland, OH 44106, USA; 5Faculty of Medicine, Normandy University, UNICAEN, ED 497, 14032 Caen, France; 6Home-Based Rehabilitation Center, FormAction Santé, 59840 Pérenchies, France; jmgrosbois@formactionsante.com

**Keywords:** acute respiratory distress syndrome, COVID-19, pulmonary rehabilitation, tracheostomy, mechanical ventilation

## Abstract

(1) Background: Intensive care unit (ICU) survivors from severe COVID-19 acute respiratory distress syndrome (CARDS) with chronic critical illness (CCI) may be considered vast resource consumers with a poor prognosis. We hypothesized that a holistic approach combining an early intensive rehabilitation with a protocol of difficult weaning would improve patient outcomes (2) Methods: A single-center retrospective study in a five-bed post-ICU weaning and intensive rehabilitation center with a dedicated fitness room specifically equipped to safely deliver physical activity sessions in frail patients with CCI. (3) Results: Among 502 CARDS patients admitted to the ICU from March 2020 to March 2022, 50 consecutive tracheostomized patients were included in the program. After a median of 39 ICU days, 25 days of rehabilitation were needed to restore patients’ autonomy (ADL, from 0 to 6; *p* < 0.001), to significantly improve their aerobic capacity (6-min walking test distance, from 0 to 253 m; *p* < 0.001) and to reduce patients’ vulnerability (frailty score, from 7 to 3; *p* < 0.001) and hospital anxiety and depression scale (HADS, from 18 to 10; *p* < 0.001). Forty-eight decannulated patients (96%) were discharged home. (4) Conclusions: A protocolized weaning strategy combined with early intensive rehabilitation in a dedicated specialized center boosted the physical and mental recovery.

## 1. Introduction

The new coronavirus SARS-CoV-2 (acute respiratory syndrome coronavirus 2) can cause a special form of severe pneumonia with acute respiratory distress syndrome (COVID-19 ARDS) that often requires prolonged mechanical ventilation, protracted sedation and paralysis in the intensive care unit (ICU) [1,2]. Survivors of a long ICU stay can develop a persistent critical illness (or chronic critical illness) and are usually described as vast resource consumers with an almost desperate prognosis [3,4,5,6]. However, to our knowledge, very little, if any, data has been published to guide the management of these chronic critically ill individuals with severe neuromuscular weakness, ventilator dependency, immunosuppression, malnutrition and an uncertain outcome. We describe a protocolized global approach to this patient population, which we call the five Ds strategy. This article describes its application to fifty very frail COVID-19 survivors, first in the ICU and then in a five-bed weaning and intensive rehabilitation center (WIRC) specifically designed to deliver intensive physical activity sessions to tracheostomized survivors still under mechanical ventilation.

## 2. Methods

### 2.1. Ethical Considerations

This single-center retrospective observational study was declared to the national commission for computing and freedom (CNIL, Commission National d’Informatique et Liberté). The data were collected in an anonymized protected electronic file. The local institutional review board (comité d’éthique médicale du centre hospitalier d’Arras) approved the study on 28 April 2022 with approval number: 2022-0401. In accordance with the French Jardé law on biomedical research, the Arras Institutional Review Board waived the need for informed consent and the approval by the national committee for the protection of the patients. All procedures were followed in accordance with the ethical standards on human experimentation and with the Helsinki Declaration of 1975.

### 2.2. Intervention

The five Ds strategy (Figure 1) was started early in the ICU and was designed as a step-by-step protocolized approach of the tracheostomized critically ill patient, in which each step is an essential prerequisite for moving on to the next one. It includes: (1) defeating delirium, deconditioning and nutritional deficiencies; (2) deventilating; (3) deflating the cuff; (4) detecting swallowing disorders and (5) decannulating. It was continued in the WIRC once the second step of the 5 Ds protocol was completed at least 6 h a day.

Tracheostomy was performed early (beyond 10 days of mechanical ventilation) to facilitate the sedation withdrawal, to improve patient’s comfort and to secure the airway access, especially when the patient was moved [7]. Patients were mobilized, taken out of bed and put in a chair as soon as possible. Anything that interfered with the patient’s mobilization was removed, which included useless catheters, central venous access and continuous infusion syringe pumps. A percutaneous gastrostomy was also performed to more actively provide the prolonged nutritional support and protein intake required for such a long-lasting resuscitation process [8].Disconnection from the ventilator was obtained only after a gradual decrease in ventilatory assistance by decreasing the inspiratory support step by step or by lengthening the periods of spontaneous ventilation [9]. Inspiratory muscle training—through repetitive short inspiratory efforts against a resistive valve—was added to shorten the duration of the ventilatory support [10]. Considering that the night is a recovery period, nocturnal ventilatory assistance was maintained until the patient was able to hold out for 12 h without the help of the ventilator for at least 2 consecutive days [11]. This step was facilitated when a thinner tracheostomy tube was used (7 mm of internal diameter, for instance), providing that the cuff was deflated and a one-way speaking valve was connected to the proximal part of the cannula. This strategy enabled the patient to go through steps 2, 3 and 4 at once and drastically shortened the weaning process from mechanical ventilation.Using a one-way speaking Passy Muir valve not only accelerates the restoration of the aerodigestive tract functions of the larynx, such as speech, cough and swallow, but it also reduces the risk of aspiration and the need for suctioning [12]. We used this strategy as soon as the patient could be disconnected from the ventilator.Tracheostomized critically ill patients often exhibit clinically undetectable episodes of aspiration (called silent aspirations) [13]. That is why a fiberoptic examination of the upper airways with a blue dye swallowing test was part of the protocol, to control the anatomical and functional integrity of the larynx.The cannula was removed when the whole step-by-step process was completed.

A physical individualized rehabilitation program was added to the five Ds protocol to boost muscular, diaphragmatic and neurocognitive recovery. Briefly, it included early sedation withdrawal, passive mobilization in bed started early in the ICU using a motorized movement therapy device and then active exercise training [7]. Patients were helped to sit on the edge of the bed, transferred to the chair and eventually put in the standing upright position with assistance. In the WIRC, each patient benefited from twice-daily exercise sessions in a special fitness room attached to the ICU and specifically equipped with the required facilities for monitoring and training vulnerable critically ill patients. Mechanical ventilatory support or high flow oxygen could be provided during physical activities in the most severe patients (See Appendix A). Skilled respiratory therapists—under the supervision of the attending intensivist—piloted the exercise sessions within the WIRC.

Given that the protocol was applied during each wave of the French COVID-19 epidemic, special considerations regarding patients’ isolation and SARS-CoV-2 were taken into account. Negative pressure was applied to the whole ICU, and patients were kept isolated until day 21. Caregivers were protected against the aerosolization of the virus with gloves, FFP3/N95 masks, googles or visors and impermeable isolation gowns. A careful disinfection of the premises and training equipment was performed after each use.

### 2.3. Evaluation Criteria

The patient’s vulnerability was evaluated by the frailty score (ranging from 1 to 9, a score >4 corresponding to a frail individual not capable of walking without assistance) [14]. Patients’ dependence was assessed by the activity of daily living score (ADL) ranging from 0 (total dependency) to 5 (maximal autonomy) [15]. Patients’ functional capacities were assessed by the maximum distance (in meters) achieved during the six-minute walking test (6′WT) carried out according to the recommendations of the American Thoracic Society [16]. It was repeated once a week, especially on the patient’s admission to the WIRC, and on the day preceding hospital discharge. The psychological status was evaluated using the hospital anxiety and depression (HAD) scale, a score >11 considered abnormally high [17].

### 2.4. Statistical Analysis

Data are expressed as median and interquartile range. Pairwise comparisons between different study times were assessed using Wilcoxon signed-rank test. Data comparisons across the three different groups were performed using Friedman’s test. The Bonferroni method was used to adjust for multiple comparisons. Statistical analysis was performed using STATA 16.0 (StataCorp LP, College Station, TX, USA). *p* < 0.05 was considered statistically significant. All reported *p*-values were 2-sided.

## 3. Results

From 20 March 2020 to 20 March 2022, 502 critically ill COVID-19 ARDS patients were managed in the ICU. Among them, 136 (27%) had developed multiple organ failure with septic shock and were tracheostomized for difficult weaning from mechanical ventilation. Eighty-one (60%) died during the resuscitation period, five were transferred to other specialized rehabilitation centers and 50 benefited from the 5 Ds protocol and were admitted to the WIRC after a median of 39 (28–58) ICU days. All of them had survived protracted multiple organ failure (maximal SOFA score 12 (10–13)) and experienced severe ICU-acquired neuromuscular weakness (worst MRC 17 ± 2), prolonged delirium, severe malnutrition and susceptibility to infections with at least one superinfection during their ICU stay. The sequences of the 5 Ds protocol-driven interventions included tracheostomy on day 12 (IQR: 9–16), gastrostomy (*n* = 44/50, 88%) on day 29 (IQR: 22–36), deventilation 12 h/24 h on day 36 (IQR: 28–54), deventilation 24 h/24 h on day 40 (IQR: 32–58) and decannulation on day 53 (IQR: 39–75) before being discharged from hospital on day 66 (IQR: 50–99). Despite continuous enteral feeding with 20–25 Kcal/kg/day, the patients exhibited severe weight loss (−10%) after the ICU experience. All the patients were more fragile and more dependent at WIRC admission compared to ICU admission (Table 1).

On arrival at the WIRC, only 18 patients (36%) were able to stand up, and all of them needed the assistance of a walker and oxygen. Two patients had to be retransferred to the ICU because of a new episode of ARDS and died during their hospital stay. The WIRC LOS was 25 (14–37) days. At the end of the rehabilitation program, patients’ autonomy (ADL) and patients’ fragility (frailty score) were significantly improved compared to patients’ functional status at the ICU discharge (Table 1). Furthermore, patients’ autonomy (ADL, median 6 (IQR: 6–6) vs. 6 (IQR: 6–6) before the ICU; *p* = 0.6) and frailty score (median 2 (IQR: 2–3) vs. 3 (IQR: 2–3); *p* = 0.4) returned to the baseline values prior to ICU admission after the rehabilitation protocol. As shown in Figure 2, the distance traveled during the six-minute walking test (6′WT) drastically increased.

At the end of the WIRC stay, less patients needed to be assisted by a walker (*n* = 7, 15% vs. *n* = 18, 100%; *p* < 0.01) and by oxygen (*n* = 15, 31% vs. *n* = 15, 83%; *p* < 0.01) compared to the first week of their arrival in the rehabilitation center. Their psychological status improved as well, with a significant decrease in the HAD score during the WIRC stay, especially on the anxiety component (Table 1).

## 4. Discussion

In this study of vulnerable survivors of CARDS with chronic critical illness, we report a surprisingly good outcome with the synergic combination of an early intensive rehabilitation program added to a protocol-driven approach of difficult weaning from mechanical ventilation.

Comparative data regarding the post-ICU burden (PICS, post-intensive care syndrome) and early rehabilitation of the COVID-19 ARDS survivors are still scarce. Before the COVID-19 epidemic, a large multi-center prospective Canadian research initiative, called the RECOVER program, studied the outcome of 391 ICU survivors who had been mechanically ventilated more than 7 days for ARDS. Using a statistical recursive partitioning model, Herridge MS et al. showed that the 7-day post-ICU functional independence measure was the best determinant of the recovery trajectory to one year after ICU discharge and impacted the one-year mortality. This relationship was independent of the admitting diagnosis causing ARDS and critical illness severity [18]. By analogy, it can thus be assumed that our early per-ICU respiratory rehabilitation program, with its significant positive effects on the aerobic capacity and mental status, might prevent post-intensive care syndrome and should improve the 1-year outcome in these COVID-19 critical care survivors. Long-term physical, cognitive and psychosocial consequences have been identified in ICU survivors from the CARDS. Several authors pleaded for the implementation of early rehabilitation in mechanically ventilated patients with CARDS [19,20,21]. Stutz et al. demonstrated the feasibility of implementing physical and occupational therapy in a population of critically ill COVID-19 patients [20]. In 43 CARDS patients admitted to a post-ICU weaning center, Faure et al. found that the MRC on admission was the only factor independently associated with early decannulation [21]. This was in accordance with our strategy favoring early exercise training to reinforce the skeletal muscles and especially the diaphragm through inspiratory muscle training. We assumed that this would accelerate the weaning process from mechanical ventilation.

Several limitations have to be acknowledged given the retrospective single-center study design. One may argue that—in the absence of a control group without rehabilitation—the results may simply reflect the natural recovery of the patients. If our population can be considered highly selected, our inclusion criteria ensured the selection of critically ill patients emerging as the most vulnerable ventilator-dependent survivors from the COVID-19 epidemic. The predicted outcome of such a frail population of tracheostomized ICU survivors may appear so poor that some may consider these weakened time-consuming individuals as better candidates for palliative care than for a futile expense of scarce resources [3,4,5,6]. The strikingly good clinical course of our tracheostomized patients questioned these misleading preconceptions and certainly could not be the result of a spontaneous natural recovery. The before-and-after study design ensures that each patient serves as his/her own control, thus limiting the confounding factors. Despite being largely advised and feasible in the ICU, early rehabilitation is too often overlooked and may be considered as a low-priority issue in the daily clinical practice of critical care [22]. Incorporating into the ICU a training room specially fit to receive chronic critically ill patients has enabled us to enforce an early intensive physical rehabilitation program in conjunction with the weaning protocol. This is truly a novel experience. Of course, this preliminary report needs to be challenged by a large multicenter randomized clinical trial. The long-term outcomes and newly acquired skills maintenance remain important questions to be explored. Herein, we shared the protocol used in our center to allow the generalization and reproducibility of our results.

This study provides preliminary evidence that an intensive rehabilitation program accelerated the physical recovery and psychological status in chronic critically ill ICU survivors from COVID-19 ARDS. A global protocolized weaning strategy, started early in the ICU and immediately followed by intensive rehabilitation in a specialized center, allowed almost all these survivors to return home with the same functional status and frailty score as before their ICU admission.

## Figures and Tables

**Figure 1 jcm-11-03921-f001:**
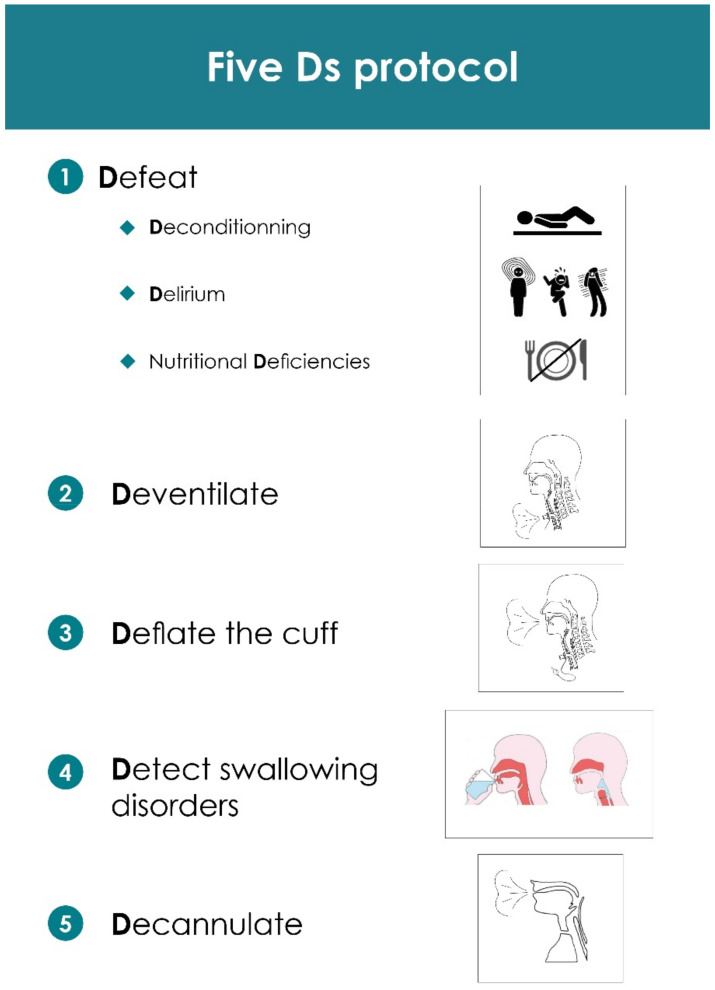
Illustration of the 5 Ds protocol up to the patients’ decannulation.

**Figure 2 jcm-11-03921-f002:**
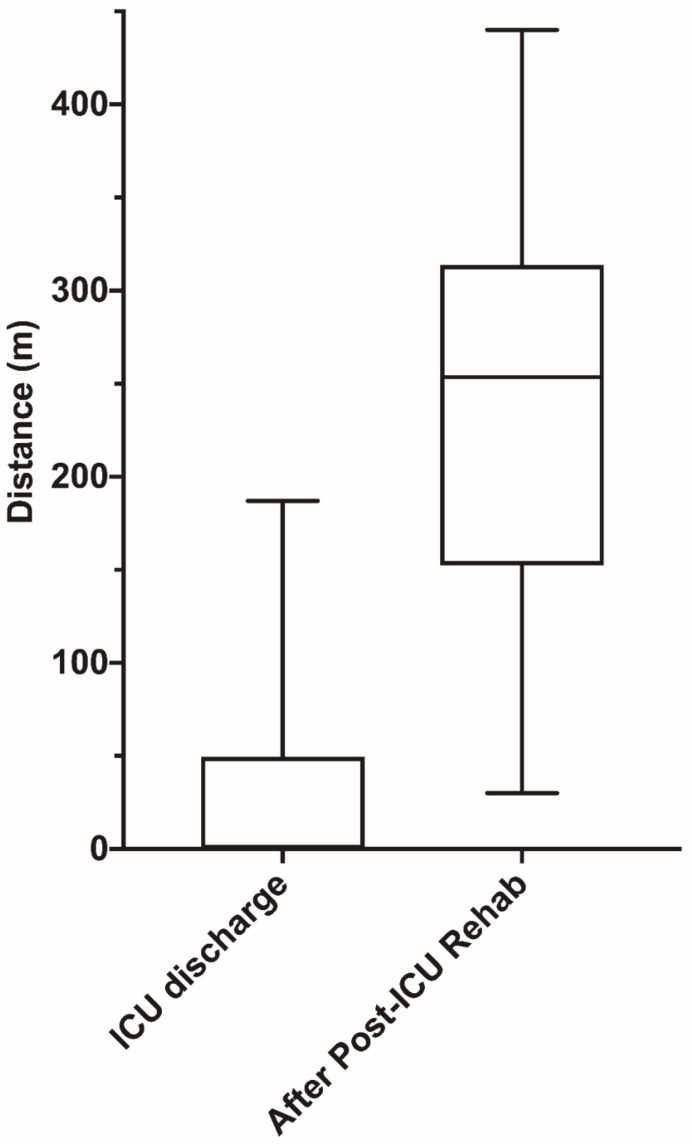
Evolution of the six-minute walking test distance between the day of ICU discharge and the last day in the post-ICU weaning and intensive rehabilitation center.

**Table 1 jcm-11-03921-t001:** Comparison of multidimensional outcome measures in 48 COVID-19 ARDS survivors between ICU admission, immediate post-ICU and the post-rehabilitation period.

Parameters	ICU Admission	Post ICU	Post Rehab	*p*
Age	66 (62–68)			
Male, *n* (%)	36 (77%)			
Weight (Kg)	88.5 (80–102)	80.5 (71–89)		<0.001
BMI (kg/m²)	31.4 (26–34)			
Frailty Score	2 (2–3)	7 (7–7)	3 (2–3)	<0.001
ADL	6 (6–6)	0 (0–0.25)	6 (6–6)	<0.001
6′WT distance (m)		0 (0–49)	253 (155–312)	<0.001
HADS		18 (14–22)	10 (7–13)	<0.001
Anxiety		12 (10–13)	7 (4–10)	<0.001
Depression		6 (5–10)	2 (1–4)	<0.001

Abbreviations: ADL, activity of daily living score; BMI, body mass index; HADS, hospital anxiety and depression scale; ICU, intensive care unit; 6′WT, six-minute walking test. Data are expressed as the median (1st–3rd quartiles). Wilcoxon signed-rank test was used to compare data between the two groups. *p* ≤ 0.05 was considered statistically significant.

## Data Availability

The data presented in this study are available on request from the corresponding author. The data are not publicly available due to the Ethics Committee restrictions.

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
