# Peer review of "Early Intensive Physical Rehabilitation Combined with a Protocolized Decannulation Process in Tracheostomized Survivors from Severe COVID-19 Pneumonia with Chronic Critical Illness"

_jcm, 2022, doi:10.3390/jcm11133921_

Round 1
Reviewer 1 Report
In this study, Lemyze et al. presented the impact of early rehabilitation performed on ICU COVID-19 survivors. The idea behind this study is excellent in terms of clinical relevance. The Authors accurately discuss the necessity of intensive rehabilitation in chronic critically ill patients with severe complications, and this topic is of major interest to physiotherapists working in hospitals.
However, I would like to see more publications in the Discussion. I understand there are not many publications about intensive rehabilitation in COVID-19 ICU survivors, but there are some articles about post-ICU PR. I think you could compare your results with existing literature and discuss your outcomes more. However, this is just a small suggestion rather than a requirement.
Overall, the manuscript is written concisely and informative, and minor improvements must be made, such as double-spacing and aligning.
Author Response
Dear Editor of Journal of Clinical Medicine,
First of all, we thank the reviewers for their careful read and thoughtful comments on previous draft. We have carefully taken their comments into consideration in preparing our revision, which, we hope, has resulted in a clearer, broader and more compelling paper. We would like to thank the editors too for giving us the opportunity to revise our manuscript.
Reviewer 1
However, I would like to see more publications in the Discussion. I understand there are not many publications about intensive rehabilitation in COVID-19 ICU survivors, but there are some articles about post-ICU PR. I think you could compare your results with existing literature and discuss your outcomes more. However, this is just a small suggestion rather than a requirement.
Response : We have added additional references (ref 19-21) and extended the discussion section of the revised version of the manuscript (p13-14 L255-263), as follows :
« Several authors pleaded for the implementation of early rehabilitation in mechanically ventilated patients with CARDS [19-21]. Stutz et al. demonstrated the feasibility of implementing physical and occupational therapy in a population of critically ill COVID-19 patients [20]. In 43 CARDS patients admitted to a post-ICU weaning center, Faure et al. found that the MRC on admission was the only factor independently associated with early decannulation [21]. This is in accordance with our strategy favoring early exercise training to reinforce the skeletal muscles and especially the diaphragm through inspiratory muscle training. We assumed that this would accelerate the weaning process from mechanical ventilation. »
Overall, the manuscript is written concisely and informative, and minor improvements must be made, such as double-spacing and aligning.
Response : We have made the corrections required by the reviewer in the revised version of the manuscript.
Kind and respectful regards.
Malcolm LEMYZE, M.D.
Department of Critical Care Medicine and Post-ICU Weaning and Intensive Rehabilitation Center, Arras Hospital, France
Reviewer 2 Report
This is an interesting study addressing Early intensive physical rehabilitation combined with a protocolized decannulation process in tracheostomized survivors from severe covid-19 pneumonia with chronic critical illness. I have a few comments as follows:
1. Since the study is retrospective in design, the confounding factors cannot be fully adjusted by simple bivariate analysis, I would like to suggest multivariable regression analysis to examine the robustness of the results.
2. A table 1 is required to compare differences between intervention versus control groups. I suggest to include a historical control before the implementation of the rehabilitation protocol.
3. without comparisons, the current conclusion cannot be supported the recovery may be attributable to the natural recovery.
4. CCI is a retrospective definition that cannot be applied on admission; so you need comment on how this study can be used in clinical practice. using the protocol for those who can recover quickly without CCI may not benefit patients.
5. The 5D protocol is actually practiced in routine clinical setting, I suspect the novelty of the study, they just invent a name for the sequential procedure.
Author Response
Dear Editor of Journal of Clinical Medicine,
First of all, we thank the reviewers for their careful read and thoughtful comments on previous draft. We have carefully taken their comments into consideration in preparing our revision, which, we hope, has resulted in a clearer, broader and more compelling paper. We would like to thank the editors too for giving us the opportunity to revise our manuscript.
Reviewer 2
This is an interesting study addressing Early intensive physical rehabilitation combined with a protocolized decannulation process in tracheostomized survivors from severe covid-19 pneumonia with chronic critical illness. I have a few comments as follows:
- Since the study is retrospective in design, the confounding factors cannot be fully adjusted by simple bivariate analysis, I would like to suggest multivariable regression analysis to examine the robustness of the results.
Response: The analysis has been performed according to a before-and-after study design, to compare the patients’ health status at the end of the ICU stay and the health status of the same population at the end of the rehabilitation program. Therefore, in the present study, each patient serves as his/her own control. Adding another population with other underlying diseases (different from the COVID-19 ARDS) would introduce much more bias, and would be counterproductive from a statistical point of view. Perhaps the reviewer may clarify what confounding factors are to be taken into account in the present study. Nevertheless, the comment of the reviewer sounds like a generic didactic criticism of retrospective studies, not so relevant for the present study.
- A table 1 is required to compare differences between intervention versus control groups. I suggest to include a historical control before the implementation of the rehabilitation protocol.
Response: Since our post-ICU rehabilitation center has been created to respond to the surge of critically ill patients with the first wave of the COVID-19 epidemic in Marsh 2020, I am afraid I cannot provide such a control group. A comparison between ICU survivors from COVID-19 ARDS and ICU survivors from non-COVID-19 ARDS would not be relevant. Moreover, it would introduce additional confounding factors. In the present study, each patient serves as his/her own control, thus limiting the confounding factors. The fact that all the patients have the same disease ─ severe COVID-19 pneumonia ─ also limits the confounding factors.
- without comparisons, the current conclusion cannot be supported the recovery may be attributable to the natural recovery.
Response: In theory, the reviewer may be right. However, according to the medical literature and to the clinical experience, such chronically critically ill patients do not spontaneously recover (Scheinhorn DJ et al. Chest 2007; Unroe M et al. Ann Intern Med 2010). All these patients had developed a severe ICU-acquired neuromuscular weakness, protracted delirium, severe malnutrition, and multiple infections, all of which made them vulnerable and ventilator-dependent. This is already mentionned in the result section of the manuscript (p9-10 L188-192) as follows:
"All of them had survived protracted multiple organ failure (maximal SOFA score 12 [10-13]) and experienced severe ICU-acquired neuromuscular weakness (worst MRC 17 ± 2), prolonged delirium, severe malnutrition, and susceptibility to infections with at least one superinfection during their ICU stay."
It is very unlikely that such frail tracheostomized ventilator-dependent survivors (from a 39 days of ICU length of stay) do recover without a specialized management. There is a considerable body of medical literature to support this point. For examples:
Scheinhorn DJ, Hassenpflug MS, Votto JJ, Chao DC, Epstein SK, Doig GS, Knight EB, Petrak RA; Ventilation Outcomes Study Group. Ventilator-dependent survivors of catastrophic illness transferred to 23 long-term care hospitals for weaning from prolonged mechanical ventilation. Chest. 2007; 131(1):76-84.
Unroe M, Kahn JM, Carson SS, Govert JA, Martinu T, Sathy SJ, Clay AS, Chia J, Gray A, Tulsky JA, Cox CE. One-year trajectories of care and resource utilization for recipients of prolonged mechanical ventilation: a cohort study. Ann Intern Med. 2010 Aug 3; 153(3):167-75.
Carson, SS, Kahn, JM, Hough, CL, et al. A multicenter mortality prediction model for patients receiving prolonged mechanical ventilation. Crit Care Med 2012; 40: 1171–1176.
Cox CE, Martinu T, Sathy SJ, et al. Expectations and outcomes of prolonged mechanical ventilation. Crit Care Med. 2009; 37(11):2888-2904.
We clarified that point in the revised version of the manuscript as follows:
"The predicted outcome of such a frail population of tracheostomized ICU-survivors may appear so poor that some may consider these weakened time-consuming individuals as better candidates for palliative care than for a futile expense of scare resources [3-6]. The strikingly good clinical course of our tracheostomized patients challenged these misleading preconceptions, and certainly cannot be the result of a spontaneous natural recovery."
- CCI is a retrospective definition that cannot be applied on admission; so you need comment on how this study can be used in clinical practice. using the protocol for those who can recover quickly without CCI may not benefit patients.
Response: Again the trajectory of care of ventilator-dependent ICU survivors with CCI is not toward spontaneous recovery. Iwashyna et al have demonstrated that critically ill patients transitioned from acute to chronic critical illness from the 10th day of mechanical ventilation. Therefrom the prognosis depends more on the preceding vulnerability and past medical history of the subject rather than on the severity on admission of the first critical insult (Iwashyna TJ et al. Lancet Respir Med. 2016; 4:566-573). In clinical practice, such patients may be early and prospectively identified, and oriented towards a specialized unit to benefit from such a protocolized rehabilitation strategy. This has been clarified in the revised manuscript as follows (p14 L265-275):
“One may argue that in the absence of a control group without rehabilitation, the results may be simply due to the natural recovery of the patients. If our population can be considered highly selected, our inclusion criteria ensured the selection of critically ill patients emerging as the most vulnerable ventilator-dependent survivors from the COVID-19 epidemic. The predicted outcome of such a frail population of tracheostomized ICU-survivors may appear so poor that some may consider these weakened time-consuming individuals as better candidates for palliative care than for a futile expense of scare resources [3-6]. The strikingly good clinical course of our tracheostomized patients challenged these misleading preconceptions, and can certainly not be the result of a spontaneous natural recovery. The before-and-after study design ensures that each patient serves as his/her own control, thus limiting the confounding factors.”
- The 5D protocol is actually practiced in routine clinical setting, I suspect the novelty of the study, they just invent a name for the sequential procedure.
Response: We have to acknowledge that the 5D’s protocol is just a mnemonic used to describe the necessary steps to overcome before the patient can be decannulated. The originality of the concept lies on the synergic association of a protocolized approach of difficult weaning combined with early exercice training in a dedicated equipped fitness room. We find it useful to help the caregivers remember the different steps that need to be achieved before the decannulation can be allowed. To meet the reviewer’s requirement, we deleted the term “new” protocolized approach in the introduction section of the revised manuscript.
Kind and respectful regards.
Malcolm LEMYZE, M.D.
Department of Critical Care Medicine and Post-ICU Weaning and Intensive Rehabilitation Center, Arras Hospital, France
Round 2
Reviewer 2 Report
My previous comments are well addressed in this updated version.